# Characteristic of Metabolic Status in Heart Failure and Its Impact in Outcome Perspective

**DOI:** 10.3390/metabo10110437

**Published:** 2020-10-29

**Authors:** Hsiang-Yu Tang, Chao-Hung Wang, Hung-Yao Ho, Jui-Fen Lin, Chi-Jen Lo, Cheng-Yu Huang, Mei-Ling Cheng

**Affiliations:** 1Metabolomics Core Laboratory, Healthy Aging Research Center, Chang Gung University, Taoyuan City 33302, Taiwan; tangshyu@gmail.com (H.-Y.T.); rflin@mail.cgu.edu.tw (J.-F.L.); chijenlo@gmail.com (C.-J.L.); chenyu7015@gmail.com (C.-Y.H.); 2Heart Failure Research Center, Division of Cardiology, Department of Internal Medicine, Chang Gung Memorial Hospital, Keelung City 20401, Taiwan; bearty@adm.cgmh.org.tw; 3Department of Medical Biotechnology and Laboratory Science, College of Medicine, Chang Gung University, Taoyuan City 33302, Taiwan; hoh01@mail.cgu.edu.tw; 4Clinical Metabolomics Core Laboratory, Chang Gung Memorial Hospital, Taoyuan City 33302, Taiwan; 5Department of Biomedical Sciences, College of Medicine, Chang Gung University, Taoyuan City 33302, Taiwan

**Keywords:** heart failure, metabolomics, BNP, phenylacetylglutamine, dimethylxanthine

## Abstract

Metabolic alterations have been documented in peripheral tissues in heart failure (HF). Outcomes might be improved by early identification of risk. However, the prognostic information offered is still far from enough. We hypothesized that plasma metabolic profiling potentially provides risk stratification for HF patients. Of 61 patients hospitalized due to acute decompensated HF, 31 developed HF-related events in one year after discharge (Event group), and the other 30 patients did not (Non-event group). The plasma collected during hospital admission was analyzed by an ultra-high performance liquid chromatography time-of-flight mass spectrometry (UPLC-TOFMS)-based metabolomic approach. The orthogonal projection to latent structure discriminant analysis (OPLS-DA) reveals that the metabolomics profile is able to distinguish between events in HF. Levels of 19 metabolites including acylcarnitines, lysophospholipids, dimethylxanthine, dimethyluric acid, tryptophan, phenylacetylglutamine, and hypoxanthine are significantly different between patients with and without event (*p* < 0.05). Established risk prediction models of event patients by using receiver operating characteristics analysis reveal that the combination of tetradecenoylcarnitine, dimethylxanthine, phenylacetylglutamine, and hypoxanthine has better discrimination than B-type natriuretic peptide (BNP) (AUC 0.871 and 0.602, respectively). These findings suggest that metabolomics-derived metabolic profiling have the potential of identifying patients with high risk of HF-related events and provide insights related to HF outcome.

## 1. Introduction

The prolongation of life expectancy in patients with cardiovascular disease is accompanied with a growing prevalence of heart failure (HF), which is recognized as a major health care problem. From a public health perspective, more than 70% of HF-related health care expenditures are explained by hospital admissions [1]. HF is a syndrome of cardiac disorder that represents the inability of heart to adequately support the circulation and the need of whole body metabolism. Regardless of advances in therapy, morbidity, and mortality of HF remain high, leading to considerable consumption of healthcare resources [2] and substantial reduction of quality of life. Much effort has been made to reduce frequent hospital readmissions, and to prevent decompensation of HF. Despite improvement in the treatment of HF, the management also remains a challenge due to limited knowledge about the pathophysiology. Outcomes might be improved by early identification of risk. However, the prognostic information offered is still far from enough. In order to evaluate patients at high risk, various risk scores have been developed to quantify the risk of re-hospitalization [3,4]. However, it remains unclear how to identify patients earlier to reduce to prevent re-hospitalization and readmission rate. In clinical, B-type natriuretic peptide (BNP), a heart hormone synthesized in atrial and ventricular cells, and N-terminal-pro hormone BNP (NT-proBNP) are released in response to changes in pressure inside the heart which can be associated with cardiac problems [5]. However, for acute decompensated heart failure, the BNP-guided therapy doesn’t improve post-discharge mortality or readmission rates [6]. Furthermore, some patients never reached the targeted BNP threshold even careful treatment. HF-associated pathophysiological changes lead to abnormalities in metabolism. Metabolite profile is dramatically altered during disease progression [7,8,9]. Metabolic alterations arising from genetic variants and environmental factors such as diet, lifestyle, and gut microbiome reflect pathological changes in an individual. Taking advantage of a combination of multiple biomarkers, metabolomics is regarded as a potential platform to identify metabolic signatures of HF patients. It is possible to identify the outcome-related metabolic profile of plasma of HF patients, regardless on the heterogeneous etiologies or medication.

In the past, research into the metabolic changes that occur during HF has primarily focused on changes in fatty acid and glucose metabolism as discussed. In normal condition, cardiomyocytes utilize fatty acids as a primary substrate to provide energy. However, with failing conditions, cardiomyocytes prefer to switch to glucose [10]. However, this hypothesis-driven process has failed to discover other metabolic pathways in heart failure. As such, untargeted metabolomics has helped to explain new underlying pathophysiological mechanisms in HF through discovery of novel metabolic pathways or drug-derived metabolites. Recently, most metabolomics studies have demonstrated metabolic alterations in blood of HF patients [11,12,13]. Changes in circulating metabolites that occur in HF reflect the whole body’s global metabolic state, that providing key insight into HF. In previous study [14], we have applied targeted metabolomics approach in combination with multiple molecules (a panel) to evaluate patients with HF. The major benefit of targeted metabolomics is enhanced sensitivity and selectivity. A variety of biomarkers for heart failure have been identified. Levels of histidine, phenylalanine, ornithine, spermine, spermidine, phosphatidylcholines significantly change at different stages of HF. Notably, metabolites including histidine, phenylalanine, spermidine, and phosphatidylcholine C34:4 significantly contributed to diagnosis of HF [14]. Of the metabolites, the physiological/endogenous metabolites are predominant, but many metabolites can be derived from exposures, microbes, and drugs [15]. A large of metabolites can be found in human circulation and response the environment or additional therapy. Moreover, the crosstalk between immune responses and different source of metabolites from gut microbiota regulate pathogens and host metabolome [16]. To study metabolite features associated with event and non-event in HF, unbiased metabolic profiling was used to study a broad perspective on changes that occur in HF and allows the discovery of previously neglected biological pathways that contribute to pathogenesis. A well-established untargeted metabolomics approach may be a useful technology for discern endogenous and exogenous metabolic differences in HF.

In this study, metabolomics profiles of the plasma from HF patients were analyzed by ultra-performance liquid chromatography time-of-flight mass spectrometry (UPLC-TOFMS). The data were subjected to multivariate analysis to identify metabolite signature distinct to the event and non-event groups which could be used to predict the outcome of HF.

## 2. Results

### 2.1. Baseline Characteristics and Laboratory Findings

A total of 61 subjects were enrolled in this study, including 30 patients in the non-event group and 31 in the event group. The definition of event is HF-related death or re-hospitalization within the first year from the initial discharge from hospital. The baseline characteristics and laboratory findings are shown in Table 1. Compared to the non-event group, the event group patients had higher levels of creatinine and B-type Natriuretic Peptide (BNP); lower levels of serum sodium and hemoglobin; and a higher prevalence of diabetes mellitus; and a higher rate of using allopurinol (a xanthine oxidase inhibitor). No significant differences between the two groups were noted in age, sex, LVEF, or other laboratory parameters.

ACEI, angiotensin-converting enzyme inhibitor; ARB, angiotensin receptor blocker; COPD, chronic obstructive pulmonary disease; Chronic kidney disease, estimated glomerular filtration rate (eGFR) < 60 mL/min/1.73 m^2^; LVEF, left ventricular ejection fraction; OHA, oral hypoglycemic agents; BNP, B-type Natriuretic Peptide. *p* value was determined by student’s *t*-test.

### 2.2. Discriminative Ability of Potential Metabolite Biomarkers for Events in HF

To examine whether plasma metabolome differs between event and non-event groups of these patients, plasma specimens from HF patients were analyzed by UPLC-TOFMS to generate an untargeted metabolome profiles. After peak alignment and removal of missing values, a total of 946 positive ions were detected in plasma. The orthogonal projection to latent structure discriminant analysis (OPLS-DA) score plots (Figure 1A) and loading plots (Figure 1B) demonstrated that metabolomics-based analysis models were able to distinguish the event group from the non-event group in HF patients. Discriminant metabolites that contributed to differences between these groups were selected according to their variable importance in the projection (VIP) scores. The OPLS-DA models were used to build up a training set based on all metabolites (Figure 1C), the Y axis-predicted scatter plots assigned samples to either event or non-event group using a priori cut-off at 0. The profile has the ability to discriminate patients with events from those without events. On the basis of the VIP score, 122 metabolites with a VIP score greater than 1 were considered to be potential biomarkers. The top 50 metabolites with significant contribution to separation between event group and non-event group are presented in Figure 1D. 

Further analysis revealed up-regulation of caffeine metabolism, amino acids catabolism, and purine metabolic pathways (Figure 2A), changes in fatty acid beta-oxidation, and down-regulation of phospholipid biosynthesis pathways in HF patients with poor outcome (Figure 2B). By Student’s *t*-test analysis, 49 of 122 metabolites were significantly different (*p* < 0.05) between the event and the non-event groups (Figure 2C). After searches against online databases (HMDB or METLIN) and in-house database, there were 19 metabolites with significantly different abundance between event and non-event groups. Acylcarnitines, including 3-methylglutarylcarnitine, butyrylcarnitine, 3-hydroxyoctanoylcarnitine, decatrienoylcarnitine, dodecenoylcarnitine, tetradecadiencarnitine, tetradecenoylcarnitine, and those related caffeine metabolism, like dimethyluric acid, dimethylxanthine, phenylacetylglutamine, were significantly higher in event group than non-event group. Lysophospholipids, like lysophsophatidylcholine (lysoPC)(14:0), lysoPC(15:0), lysoPC(16:0), lysoPC(18:2), LysoPC(20:5), lysophosphatidylethanolamine (lysoPE)(18:2), lysophosphatidylserine (lysoPS)(18:1), and other metabolites, like tryptophan and hypoxanthine, were significantly lower in event group than non-event group (Table 2). To understand relation between those metabolites and BNP level, a Pearson’s correlation analysis was performed to analyze the correlation between the level of BNP and differentially abundant metabolites (Figure 2D). Acylcarnitines and phenylacetylglutamine positively correlate with BNP, and lysoPCs like lysoPC(14:0), lysoPC(15:0) negatively correlate with BNP level.

The identified metabolites were comparable to that of BNP in discrimination between event and non-event groups. Each of 19 metabolites and their combinations were added to established risk prediction models of event patients and receiver operating characteristics (ROC) analysis (Appendix A). Combination of four metabolites (tetradecenoylcarnitine + dimethylxanthine + phenylacetylglutamine + hypoxanthine) allows a better prediction for event in HF than BNP (area under the curve (AUC) were 0.871 and 0.602, respectively) (Figure 3A). The DeLong’s test showed a significant difference (*p* < 0.01) between four metabolites and BNP. Comparison of BNP with four-metabolite panel showed a significantly higher predictive power by the integrated discrimination improvement (IDI) and the net reclassification improvement (NRI) (Figure 3C). When the 4-metabolite panel was combined with BNP for ROC curve analysis (AUC = 0.874), the DeLong’s test showed no significant difference (*p* = 0.502) as compared to 4-metabolite panel. In addition, we used the same model to predict the event even death in HF (Figure 3B), 4-metabolite panel (tetradecenoylcarnitine + dimethylxanthine + phenylacetylglutamine + hypoxanthine) had good discrimination for death (AUC = 0.967). The 4-metabolite panel showed a significantly higher ability of prediction by IDI and NRI (Figure 3D). However, the combination of four metabolites with BNP had achieved the highest ROC AUC 0.988, but the Delong’s test showed no significant difference between 4-metabolite and 4-metabolite plus BNP panels. The results of univariate ROC curve analyses indicated that tetradecenoylcarnitine, dimethylxanthine, phenylacetylglutamine, and hypoxanthine in the plasma samples can be good predictor for event in heart failure.

## 3. Discussion

Our study demonstrated that untargeted metabolomics analysis of plasma metabolites is able to separate HF patients with and without events. Untargeted metabolomics approaches lead to discovery of biomarkers that may not be observed in targeted biochemical assays. The outcome-associated disturbances in metabolism are involved in a number of biochemical pathways including fatty acid beta-oxidation, amino acids, phospholipid biosynthesis, caffeine and purine metabolism (Figure 2B) which may be affected by cellular endogenous, ingestions, drugs, or derived from gut microbiota. Identification of metabolites by untargeted metabolomics approach could help us to construct the overall molecular mechanism in heart failure. In the present study, we found that endogenous as well as exogenous metabolites, like phenylacetylglutamine and dimethylxanthine, could be as potential markers to assist prediction in event of heart failure. The liver is the principal location of lysophophatidylcholines and xenobiotic metabolism after ingestion. Plasma metabolome of heart failure is indicative of interactions between various organs and heart during the HF progression.

The pathogenic role of plasma lipid abnormalities in cardiovascular disease, diabetes, and kidney disease is well recognized. In our previous study, phosphatidylcholines levels were significantly lower in stage C of HF than control group [14]. Consistent with this, the lysophosphatidylcholine levels in heart failure patients with events were lowered (Table 2). However, finding from recent clinical studies of lysoPC have been controversial. Lysophosphatiditylcholine consists of a glycerin frame, a phosphocholine, and a fatty acid. It is a major component of oxidized LDL and increases lysoPC content by phospholipase A2 hydrolysis [17]. Lysophoshatidylcholines are increasingly important as key markers associated with cardiovascular diseases [18]. Herein, the present study showed the lysoPC levels decreased in HF patients with events, Lee et al. [19] also demonstrated that low lysoPC was associated with significantly higher risk of cardiovascular disease. But the underlying mechanisms for this phenomenon is not clear. Previous studies reported that the plasma lysoPC levels are reduced in patients with impaired glucose tolerance [20], and those with obesity and type 2 diabetes [21]. The mechanisms responsible for the reduction in circulating lysoPC are likely to involve reduction in lysoPC biosynthesis and/or enhanced breakdown of lysoPC and rapid clearance from the circulation by metabolically active tissues. For instance, phosphatidylcholine production from liver may decline due to liver congestion associated with HF [22]. The activity of phospholipase A2 and the serum enzyme lecithin-cholesterol acyltransferase may be affected in a severe disease status such as HF [22].

Acylcarnitines represent by-products of mitochondrial beta-oxidation. Unlike acyl-CoAs, which cannot cross the mitochondrial membrane, the acylcarnitines pass efficiently into the cytosol, and subsequently into the bloodstream. In recent studies, higher levels of short- to medium-chain dicarboxylacylcarnitines are associated with increased risk of death among patients with coronary artery disease [23]. In our result, we found the levels of 3-methylglutarylcarnitine, a dicarboxylacylcarnitine, and 3-hydroxyoctanoylcarnitine significantly are higher in the event of heart failure. Fatty acids are first converted into hydroxymonocarboxylic acid by cytochrome p450 (CYP) in microsome and then converted to dicarboxyl acids [23], and are shortened via β-oxidation in peroxisome in the liver. However, the associations between circulating acylcarnitines with various chain lengths have not been evaluated. The esterification of carnitine is catalyzed by acyltransferases specific for fatty acids of various chain lengths [24]. Accumulation of C4- to C12-acylcarnitines in patients with event can be attributed to defective β-oxidation of medium- and short chain fatty acids and /or to defective coupling between fatty acid oxidation and electron transport chain [25]. C4- to C12-acylcarnitines were associated with levels of BNP, a disease severity marker (Figure 2D), which was consistent with previous studies [14]. The incomplete fatty acid β-oxidation downstream of CPT-I lead to accumulation of acyl-CoAs in mitochondrial matrix and are converted into acylcarnitines. They are subsequently transported through mitochondrial membranes to cytosol and then to blood [26]. These findings suggest that HF patients with event have severe mitochondrial dysfunction and an abnormal acylcarnitine profile.

The phenylacetylglutamine can be a predictor for event in heart failure (Figure 3). Phenylacetylglutamine is derived from glutamine and phenylacetyl-CoA in phenylalanine transaminase-catalyzed reaction in phenylalanine metabolic pathway. It is a major nitrogenous metabolite that accumulates in the patients with congenital phenylketonuria [27] and with disorders related to urea cycle dysfunction [28]. Phenylacetylglutamine formation is considered as a mechanism to reduce whole body ammonia level through reaction with glutamine, and its accumulation may be associated with increased amino acid degradation. Its increase in HF patients with event implies that these patients use more amino acids as energy source and produce more plasma phenylacetylglutamine than those without event. Recently, phenylacetylglutamine is identified as microbial metabolites from amino acid fermentation [1]. Phenylacetylglutamine is not significantly bound to plasma proteins, the clearances of phenylacetylglutamine is highly efficient [1]. In individuals with normal renal function, the clean rate of phenylacetylglutamine is approximately 3-fold higher than creatinine clearance. The high level of plasma phenylacetylglutamine is a strong and independent risk factor for mortality and cardiovascular disease [29], revealing an imbalance between microbial metabolism and renal tubular secretion [30]. These findings suggest that eGFR was not different between event and non-event group, and the accumulation of phenylacetylglutamine may due to the increase in amino acid metabolism.

Levels of dimethylxanthine and dimethyluric acid significantly increased in event patients than non-event patients, indicating the disturbance of caffeine metabolism pathway. Dimethylxanthines, including three isomers of theophylline (1,3-dimethylxanthine), theobromine (3,7-dimethylxanthine), and paraxanthine (1,7-dimethylxanthine), are metabolized from caffeine. Each isomer has its characteristic pharmacological effects. In our result, the dimethylxanthines are significantly increased in event group versus non-event group. To evaluate which isomer is involved, we performed the UPLC-QTOFMS with charged surface hybrid C18 column (Waters Corp; Milford, MA, USA) to separate those isomer (Appendix A). Theophylline and paraxanthine levels significantly increase in the event group (Appendix A), and theophylline contribute most abundance. Theophylline, an isomer of dimethylxanthines, is often used in the asthma and chronic obstructive pulmonary disease (COPD) [31,32]. Several studies have reported the side effects of theophylline, which include immunomodulation [33] and cardiac arrhythmias at high concentrations [31,34]. The rate of theophylline metabolism and clearance vary among individuals, resulting in different serum drug concentrations that affects the drug efficacy and treatment dosage. For instance, disease can induce epigenetic changes that potentially affect enzyme activity and capacity of metabolizing enzymes, like cytochrome P450 (CYP) superfamily [35], and the drug efficacy and toxicity. Inter-individual variability in expression and function of CYP enzymes is a major factor accounting for individual susceptibility to drug response. The ability of CYP enzyme ability to metabolize arachidonic acid to epoxyeicosatrienoic acids and hydroxyeicosatetraenoic acids is important in maintenance of the cardiovascular health [36]. It is well documented that expression and activities of CYP enzymes are downregulated in heart failure [37], which leads to decline of the efficacy of drug clearance. The administration of theophylline and caffeine has been reported to increase plasma renin activity [38]. This effect is also observed in patients with congestive HF. Theophylline acts as an antagonist and reduces the inhibitory effect of adenosine, resulting in a more than two-fold increase in plasma renin activity in CHF [39]. Moreover, significant differences were observed in creatinine levels in patients in the event group, but kidney function and caffeine metabolite elimination were not observed.

Changes in xanthine oxidation pathway and uric acid itself play important pathophysiological roles in HF [40,41]. Clinically, decline in uric acid level through xanthine oxidase inhibition may improve endothelial reactivity [42], myocardial function [43], and ejection fraction [43,44], and is associated with better outcomes in HF patients [10,40]. Not only do clinical studies support the direct involvement of xanthine oxidation pathway in HF progression, but they also suggest xanthine oxidase as novel therapeutic target and better therapy, which potentially leads to improve myocardial efficiency [43,44]. Compared to the non-event group, the event group had a significant decrease in hypoxanthine, and a trend of increase in xanthine. This remains to be clarified whether these changes are associated with the use of some xanthine oxidase inhibitors and medication such as allopurinol and benzbromazone to increase uric acid excretion.

BNP level has been used as a routine biomarker for diagnosis congestive HF. BNP is a family of protein hormones called natriuretic peptides produced by muscle cells. As the heart chambers are dilated, heart muscle cells are stretched to accommodate extra blood leading to production of BNP. BNP is excellent for clinical diagnosis. However, BNP does not usually cause immediate changes in normal heart because its synthesis and secretion increases in response to hemodynamic stimuli over a long time [17]. The BNP level is closely related to the incidence and severity of HF which plays a crucial role in the process of therapy [45]. However, for acute decompensated heart failure, the BNP-guided therapy doesn’t improve post-discharge mortality or readmission rates [6]. Furthermore, some patients never reached the targeted BNP threshold even careful treatment. In our result, the predictive power of BNP to incidence of readmission and death is AUC 0.602 and AUC 0.638 (Figure 3). The combination of four metabolites (tetradecenoylcarnitine + dimethylxanthine + phenylacetylglutamine + hypoxanthine) the power increase to 0.871 and 0.967. Apparently, the combination of 4 metabolites serves as a better predictor for prognosis of HF event than BNP. This probably prevents high mortality associated with relapse of disease.

The results of metabolite profile in present study has verified the high predictive significance in event of HF by incorporating tetradecenoylcarnitine, dimethylxanthine, phenylacetylglutamine, and hypoxanthine. However, there are some limitations in this study. The small sample size may lower the statistical power. Nevertheless, we have clearly shown differences in metabolic profiles between event and non-event HF patients. The prognostic power of metabolites in HF needs to be addressed by increasing the population in future investigations. On the other hand, some unknown interactions of nutrition and medications may partially contribute to the metabolic differences between groups as well. As shown in Table 1, there were no significant differences in medications used for HF between the two groups.

## 4. Materials and Methods

### 4.1. Patients and Study Design

The study enrolled patients consecutively hospitalized for acute or decompensated chronic HF in the HF center of Chang Gung Memorial Hospital, Keelung, Taiwan. Enrollment criteria included patients (1) with typical signs and symptoms of HF, the severity of which fits New York Heart Association (NYHA) functional classification III to IV; who were hospitalized due to acute cardiogenic pulmonary congestion based on chest x-rays and non-cardiogenic cases were excluded; (2) with structural abnormalities documented by echocardiograms and with LVEF < 50%; and (3) between 39 and 80 years of age. Exclusion criteria included (1) the presence of systemic diseases such as hypothyroidism, decompensated liver cirrhosis, and systemic lupus erythematosus; (2) the presence of disorder other than HF that might compromise survival within 6 months; (3) patients being bed-ridden for >3 months and/or unable to stand alone; (4) patients with serum creatinine of >3 mg/dl; and (5) patients with severe coronary artery disease without complete revascularization therapy. Informed consent was obtained from all patients. The study was designed and carried out in accordance with the principles of the Declaration of Helsinki and with approval from the Ethics Review Board of Chang Gung Memorial Hospital.

Patients were followed up for one year. Follow-up data were prospectively obtained every month from hospital records, personal communication with the patients’ physicians, telephone interviews, and patients’ regular visits to staff physician outpatient clinics. Events of HF-related death or re-hospitalization were recorded. A committee of three cardiologists, who were blinded to patients’ clinical outcomes, adjudicated all hospitalizations to determine whether the events were related to worsening HF. Patients were followed for 1 year. In those developing events including HF-related death or re-hospitalization in the first year, the 31 patients were defined as the “event group”. In those without HF-related events in the first year, the 30 patients were defined as the “non-event group”.

### 4.2. Blood Samples Collection

Blood samples were collected in EDTA-containing tubes. Plasma was analyzed by metabolomic workflow described in the succeeding section. BNP was measured in triplicate with the Triage BNP Test (Biosite, San Diego, CA, USA), which was a fluorescence immunoassay for quantitative determination of plasma BNP. Precision, analytical sensitivity, and stability characteristics of the assay were previously described [46]. Kidney function, hemoglobin, and C-reactive protein were measured in the central core laboratory of Chang Gung Memorial Hospital.

### 4.3. Metabolomics Analysis by UPLC-TOFMS

To 50 μL plasma, 200 μL acetonitrile was added. The mixture was vortexed for 30 s; sonicated for 15 min; and centrifuged at 10,000× *g* for 25 min. The supernatant was collected in a separate glass tube. The pellet was re-extracted once with 200 μL of 50% methanol. The methanolic and acetonitrile phases were pooled and dried in a nitrogen evaporator. Residue was suspended in 100 μL of 95:5 water:acetonitrile; and centrifuged at 14,000× *g* for 30 min. Liquid chromatographic separation was performed on a 100 mm × 2.1 mm, Acquity 1.7 μm C8 column (Waters Corp., Milford, MA, USA) using a ACQUITY TM UPLC system (Waters Corp., Milford, MA, USA). The column was maintained at 45 °C, and at a flow rate of 0.5 mL/min. Analytes were eluted from LC column using with a linear gradient: 0–2.5 min: 1–48% B; 2.5–3 min: 48–98% B; 3–4.2 min: 98% B; 4.3–6 min: 1% B for re-equilibration. The mobile phase were 0.1% formic acid in water (Solvent A) and 0.1% formic acid in acetonitrile (solvent B). The eluent was introduced into the Synapt G1 high-definition mass spectrometer (Waters Corp., Milford, MA, USA) operated in positive ion mode. The conditions were as follows: desolvation gas was set to 700 L/h at a temperature of 300 °C, the cone gas set to 25 L/h and the source temperature set at 80 °C. The capillary voltage and cone voltage were set to 3000 and 35 V, respectively. The data were collected in centroid mode from 20 to 1000 *m*/*z*. The peak finding, filtering, and alignment from raw mass spectrometric data were performed by MarkerLynx 4.1 software (Waters Corp., Milford, MA, USA).

Exact molecular mass data for metabolites, which showed significant differences between two groups, were then submitted for database searching, either using in-house database or online HMDB (http://www.hmdb.ca/) or METLIN (metlin.scripps.edu/index.php) database. For identification of specific metabolites, standards were subject to UPLC-TOFMS analyses under the conditions identical to those of the profiling experiment.

### 4.4. Statistical Analyses

To maximize the identification of differences in metabolic profiles between groups, the orthogonal projection to latent structure discriminant analysis (OPLS-DA) model was applied and performed using the SIMCA-P software (version 13.0, Umetrics AB, Umea, Sweden). The variable importance in the projection (VIP) value of each variable in the model was calculated to indicate its contribution to the classification. A higher VIP value represents a stronger contribution to discrimination between groups. The VIP values of those variables greater than 1 are considered to have high contribution in difference. Results are expressed as the mean ± SD for continuous variables and as the number (percentage) for categorical variables. Data were compared by two-sample *t*-tests. A *p* value of <0.05 was considered statistically significant. FDR corrections were performed for comparisons between groups where appropriate (q-value). The correlation coefficient analysis was performed by MetaboAnalyst 4.0. The receiver operating characteristic (ROC) curves were assessed to predict the event in heart failure. The comparison between the area under curve was analyzed using DeLong’s test. Improvement in the ability of prediction in different models was assessed using the IDI and NRI in the logistic regression model. All statistical analyses were performed using IBM SPSS 20.0 (Armonk, NY) and R V. 4.0.2 (R Development Core Team).

## Figures and Tables

**Figure 1 metabolites-10-00437-f001:**
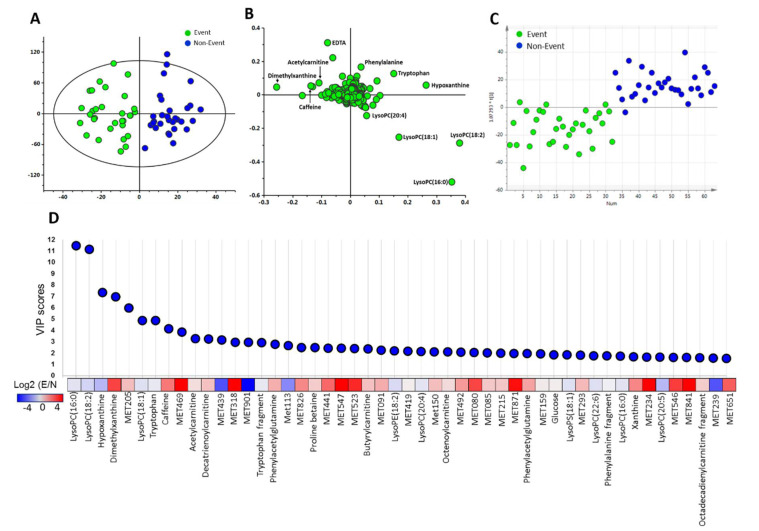
Changes in metabolome of heart failure patients. The OPLS-DA score plot (**A**) show considerable separation between 30 non-event patients (blue) and 31 heart failure (HF) event patients (green). The ellipse shown in the model represents the Hotelling T2 with 95% confidence. Each point represents one patient, and the cohort to which they belong is color-coded as shown in the legend box. The loading plot (**B**) of metabolite profiles, each point in represents one metabolite. (**C**) Score plots of orthogonal projection to latent structure discriminant analysis (OPLS-DA) prediction model is used to determine the success of the models for classifying the event and non-event groups. All of the sample were applied to construct the model, and the Y axis-predicted scatter plots assigned samples to either event or non-event group using a priori cut-off at 0. (**D**) The top 50 metabolites were ranked according to variable importance in the projection (VIP) score of OPLS-DA model. The heat map shows the ratio of abundance (log2) between event and non-event group. The un-identified metabolites are showed with metabolite ID.

**Figure 2 metabolites-10-00437-f002:**
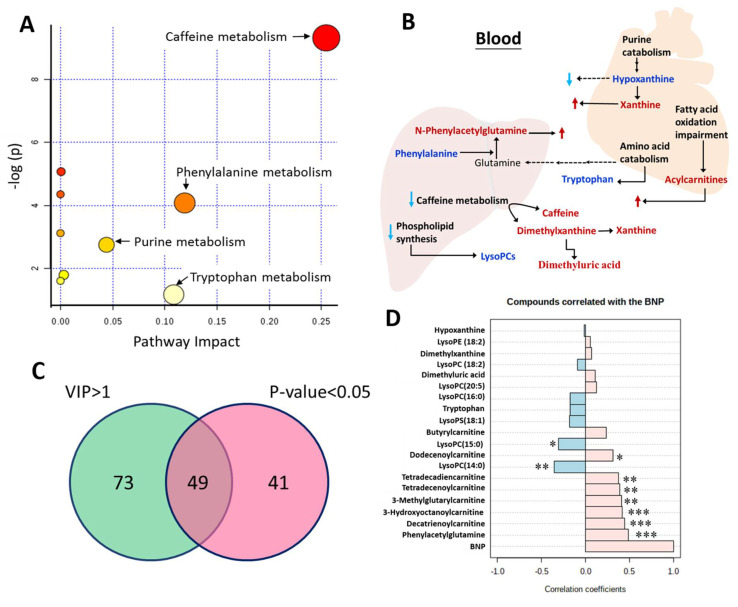
Identification of metabolites between event and non-event groups. (**A**) Pathway analysis reveals that several pathways were changed in HF patients with event group. Metabolic disturbances are mapped to pathways involved in amino acid metabolism, caffeine metabolism, and purine metabolism. (**B**) Scheme of metabolic disturbance associated with poor heart failure-related outcomes. Red metabolites: those significantly increasing in event group; blue metabolite: those significantly decreasing in event group; dark metabolites: not measured or no significant difference between event and non-event groups. (**C**) Venn diagram shows the number of metabolites with VIP greater than 1 and significantly (*p* < 0.05) altered between event and non-event groups. (**D**) Significantly different metabolites were correlated with level of BNP by Pearson’s correlation coefficient analysis. * *p* < 0.05, ** *p* < 0.01, *** *p* < 0.001.

**Figure 3 metabolites-10-00437-f003:**
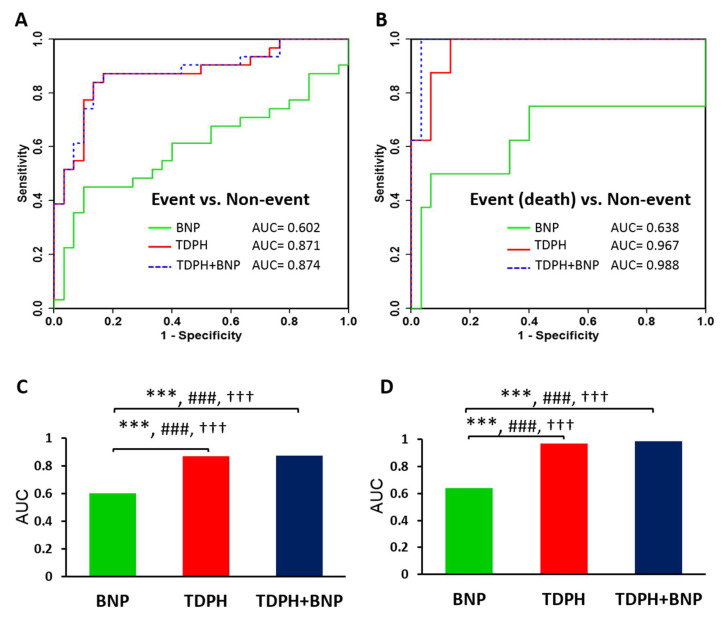
ROC AUC analysis plots. Comparison between the AUC with (**A**) ROC curves for the discrimination of the event patients and non-event patients using BNP, four metabolites (tetradecenoylcarnitine + dimethylxanthine + phenylacetylglutamine + hypoxanthine (TDPH)). (**B**) ROC curves for the discrimination of the event patients (death) and non-event patients using BNP, four metabolites (tetradecenoylcarnitine + dimethylxanthine + phenylacetylglutamine + hypoxanthine (TDPH)). The AUC values for respective ROCs in event versus non-event (**A**,**C**) and event (death) versus non-event (**B**,**D**) models are shown. Statistically significant differences between two AUCs with DeLong’s test are indicated *** *p* < 0.001. Significant increases in ability of prediction as assessed by integrated discrimination improvement (IDI) and the net reclassification improvement (NDI) are indicated # and †, respectively. ### or ††† *p* < 0.001.

**Table 1 metabolites-10-00437-t001:** Characteristics and laboratory data of 61 heart failure patients in event and non-event group.

Variable	Non-Event (*n* = 30)	Event (*n* = 31)	*p*-Value
Age (years)	60.3 ± 11.7	64.5 ± 11.6	0.173
Male (%)	24 (80.0)	20 (64.5)	0.416
EF (%)	32.2 ± 10.4	34.2 ± 13.7	0.518
Blood pressure (mm Hg)			
Systolic	120.2 ± 17.5	121.9 ± 16.5	0.703
Diastolic	75.2 ± 13.2	70.0 ± 10.9	0.094
Heart rate (beats/min)	76.9 ± 10.1	79.5 ± 11.5	0.338
Co-morbidity			
Diabetes mellitus (%)	6 (20.0)	15 (48.4)	0.031
Hypertension (%)	20 (66.7)	23 (74.2)	0.582
Atrial fibrillation (%)	4 (13.3)	10 (32.3)	0.127
COPD (%)	2 (6.7)	6 (19.4)	0.255
Ischemic (%)	19 (63.3)	16 (51.6)	0.440
Body mass index (kg/m^2^)	24.4 ± 4.9	24.8 ± 4.9	0.771
Medication			
ACEI or ARB (%)	28 (93.3)	24 (77.4)	0.147
β-Blocker (%)	27 (90.0)	21 (67.7)	0.06
Diuretic (%)	23 (76.7)	23 (74.2)	1.000
Allopurinol	0 (0)	6 (19.4)	0.024
Benzbromarone	0 (0)	2 (6.5)	0.492
Laboratory data			
Serum sodium (mEq/L)	140.2 ± 2.3	137.6 ± 4.7	0.008
Hemoglobin (g/dL)	14.3 ± 1.9	13.0 ± 2.4	0.028
Total bilirubin (mg/dL)	1.1 ± 0.6	1.3 ± 0.8	0.259
Albumin (g/dL)	3.7 ± 0.5	3.5 ± 0.6	0.169
eGFR (ml/min/1.73 m^2^)	74.3 ± 18.6	62.8 ± 37.3	0.134
Creatinine (mg/dL)	1.0 ± 0.3	1.5 ± 1.1	0.024
Cholesterol (mg/dL)	171.7 ± 44.5	173.1 ± 63.1	0.919
TG (mg/dL)	111.6 ± 48.2	125.7 ± 67.3	0.351
LDL-cholesterol (mg/dL)	111.1 ± 41.7	115.3 ± 60.1	0.755
HDL-cholesterol (mg/dL)	36.4 ± 10.6	34.7 ± 15.4	0.615
HbA1c (%)	6.3 ± 1.4	6.8 ± 1.7	0.234
BNP (pg/mL)	518.2 ± 620.9	891.5 ± 882.6	0.063
CRP (mg/L)	19.3 ± 43.4	16.4 ± 25.8	0.758
Uric acid (mg/dL)	7.4 ± 2.1	8.1 ± 2.9	0.262

**Table 2 metabolites-10-00437-t002:** Statistical analysis of the metabolites differently abundant in event versus non-event groups (VIP score of >1.0 and *p* value < 0.05).

Compound Name	Retention Time (min)	m/z	Non-Event (*n* = 30)	Event (*n* = 31)	*p*-Value	VIP Score	q-Value
**Acylcarnitines**							
3-Methylglutarylcarnitine	0.95	290.1592	0.8 ± 0.9	3.4 ± 5.4	0.013	1.29	0.065
Butyrylcarnitine	1.01	232.1542	9.8 ± 5.7	16.4 ± 12.7	0.011	2.08	0.064
3-Hydroxyoctanoylcarnitine	1.33	304.2118	5.6 ± 4.8	9.1 ± 7.4	0.034	1.38	0.108
Decatrienoylcarnitine	1.54	310.2016	17.0 ± 11.9	30.9 ± 21.7	0.003	3.23	0.048
Dodecenoylcarnitine	1.85	342.2636	3.6 ± 3.3	6.7 ± 6.7	0.026	1.34	0.104
Tetradecadiencarnitine	1.92	368.2797	3.7 ± 3.6	6.3 ± 5.8	0.043	1.16	0.119
Tetradecenoylcarnitine	2.00	370.2951	2.7 ± 1.9	4.9 ± 4.2	0.009	1.23	0.055
**Lysophospholipids**							
LysoPC(14:0)	2.28	468.3087	12.4 ± 6.0	9.1 ± 6.7	0.049	1.29	0.122
LysoPC(15:0)	2.39	482.3248	9.0 ± 4.1	6.9 ± 3.9	0.044	1.05	0.116
LysoPE(18:2)	2.42	478.2926	35.5 ± 14.1	26.9 ± 17.2	0.037	2.15	0.116
LysoPC(18:2)	2.43	520.3401	590 ± 252	413 ± 236	0.006	11.11	0.050
LysoPC(20:5)	2.43	542.3244	7.3 ± 6.6	4.0 ± 4.4	0.027	1.37	0.103
LysoPS(18:1)	2.49	524.2956	31.6 ± 11.2	25.4 ± 11.9	0.040	1.81	0.115
LysoPC(16:0)	2.49	518.3259	33.7 ± 10.2	28.1 ± 10.7	0.040	1.73	0.138
**Other metabolites**							
Hypoxanthine	0.65	137.0459	174 ± 17	93.3 ± 81.4	0.023	6.93	0.102
Dimethyluric acid	0.96	197.0669	0.2 ± 0.7	3.3 ± 6.7	0.014	1.41	0.070
Dimethylxanthine	1.05	181.0717	11.5 ± 24.6	87.7 ± 143.9	0.007	7.30	0.045
Tryptophan	1.06	205.0979	19.4 ± 4.0	16.9 ± 4.9	0.027	1.21	0.100
Phenylacetylglutamine	1.25	265.1184	3.1 ± 2.2	8.2 ± 8.9	0.004	1.93	0.051

Data are expresses as Mean ± SD. LysoPC: lysophosphatidylcholine; LysoPE: lysophosphatidylethanolamine; Metabolites were identified by ms/ms and m/z values of fragments were compared with those available from online HMDB (http://www.hmdb.ca/) and in-house database. Q-value is a *p*-value which were adjusted for the false discovery rate (FDR).

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
