# Peer review of "Characteristic of Metabolic Status in Heart Failure and Its Impact in Outcome Perspective"

_metabolites, 2020, doi:10.3390/metabo10110437_

Round 1
Reviewer 1 Report
This revised manuscript by Tnag et al. focused on the predictive ability of metabolomics-derived metabolic profile for new-onset heart failure (HF) compared with that of BNP. Authors reanalyzed data and demonstrated that the area under ROC curve (AUC) of the combination of 4 metabolites, including tetradecenoylcarnitine, dimethylxanthine, phenylacetylglutamine, and hypoxanthine, for new-onset HF and death was larger than of BNP. Authors revised the manuscript according to most of reviewers’ comments. Consequently, the results were slightly different from original version. Therefore, authors should interpret the results precisely based on this version.
Authors may want to reconsider several issues as follows.
Major comments;
1) In this revised version, authors changed to combination metabolites from 2 to 4, and compared AUCs between the groups statistically. Consequently, AUC of the combination of 4 metabolites was larger than that of BNP, whereas AUC of the combination of 4 metabolites + BNP was comparable with that of the combination of 4 metabolites only. Therefore, the last sentence in abstract that “And the combination of BNP with tetradecenoylcarnitine, dimethylxanthine, phenylacetylglutamine, and hypoxanthine has excellent predict power for event patients.” is wrong. Due to the same reason, the last paragraph in the Discussion is also misinterpretation of the present results. Authors cannot conclude that predictive ability of the combination of 4 metabolites + BNP for events is superior to that of the combination of 4 metabolites only from the present results. Authors should interpret the results precisely based on this version and revise it correctly.
Author Response
Response: We are thankful to your suggestion. We have corrected the conclusion according to our result in Abstract and last paragraph in Discussion.
Reviewer 2 Report
Heart failure (HF) absorbs a significant amount of human and economic resources. HF has been known as a systemic, multi-organ syndrome with metabolic failure the basic mechanism. In this study, the authors analyzed metabolomics profiles in plasma of patients with HF using UPLC-TOFMS method combined with OPLS-DA model and found that BNP, tetradecenoylcarnitine, dimethylxanthine, phenylacetylglutamine and hypoxanthine are involved in prediction for event in HF. This study indicated that the metabolomics-derived metabolic profiling acts as a potential strategy to predict the outcome of HF.
Comments:
- BNP has been a parameter to diagnose heart failure. However, in this study, the BNP was not significant difference in event and non-event groups in Table 1. Is it because of small sample size in this study or some reasons else? Please explain it.
- As table 2 shown, lysophosphatidylcholines (LPC) were decreased in event group, compared with non-event group. Authors discussed that 1) changed LPC was associated with higher risk of cardiovascular disease; 2) LPC functions in lipid metabolism. Do authors find any differences in physiological phenotypes of lipid metabolism in patients with HF, such as LDL/HDL ratio or oxidized LDL?
- Please add literature in line 244-245 “phosphatidylcholine production from liver may decline due to liver congestion associated with HF”.
- Please clarify the legend of figure 3.
- These are some typos:
Line 117: student t-test --> student’s t-test;
Line 244: metabolism-active tissues --> metabolically active tissues;
Line 247: represent byproducts --> represent by products; (space between by and products)
Line 255: in liver --> in the liver;
Line 385: water (Solvent A) and in acetonitrile (solvent B), either Solvent A and Solvent B or solvent A and solvent B.
Round 2
Reviewer 1 Report
This revised manuscript by Tang et al. focused on the predictive ability of metabolomics-derived metabolic profile for new-onset heart failure (HF) compared with that of BNP. Authors revised the manuscript according to the reviewers’ comments. It appeared better.
Major comments;
None
Minor comments;
1) Authors should prepare a file of “Response to Reviewer”, not only revised manuscript.
This manuscript is a resubmission of an earlier submission. The following is a list of the peer review reports and author responses from that submission.
Round 1
Reviewer 1 Report
This manuscript by Tnag et al. focused on the predictive ability of metabolomics-derived metabolic profile for new-onset heart failure (HF) compared with that of BNP. Authors demonstrated that the area under ROC curve (AUC) of 2 metabolites including decatrienoylcarnitine and phenylacetylglutamine for new-onset HF and death was larger than of BNP. I agree that the seeking of better predictive marker for clinical events than traditional markers is important. Therefore, the concept of this study is understandable and the results seem reasonable. However, authors did not show any statistical evidence that superiority or improvement of predictive ability in the present study. Authors may want to consider several issues as follows.
Major comments;
1) If authors want to show the superiority of predictive ability of 2 metabolites than BNP, AUCs between 2 parameters should be compared statistically using statistical method such as DeLong’s test. Although the AUC of 2 metabolites looks better than that of BNP, there is no evidence of the deference between them in the present study.
2) If authors want to show the superiority of predictive ability of a combined parameter of 2 metabolites + BNP than 2 metabolites alone or BNP alone, authors may want to consider to calculate the net reclassification improvement (NRI) and the integrated discrimination improvement (IDI) in addition to the simple comparison of 2 parameters. Otherwise, there is no evidence of the improvement of predictive ability when 2 metabolites were added to BNP.
Minor comments;
1) Abbreviations of UPLC-TOFMS and BNP should be spelt out even in abstract.
2) Abbreviations of UPLC-TOFMS in abstract, UPLC-Q-TOFMS, UPLC-QTOF/MS, LC-QTOF MS, UPLC-MS/MS in main text should unified. Are these abbreviations the same term?
3) It seems better to describe values of mean and standard deviation with the first decimal place.
4) In Figure 3, what means 3 metabolites? If it is “2 metabolites + BNP”, please describe so, because BNP is not a metabolite.
5) Number of patients are small to investigate your aim.
6) The term of “event” used in group name may mislead as any clinical event in patients with HF. If the definition of event was new-onset HF, it seems better to describe so.
7) If predictive ability of metabolites was really superior to that of BNP, they may not be useful for clinical situation since metabolites used in this study are not easy to measure in general clinical setting.
Reviewer 2 Report
The authors should describe how identification of High risk HF patients based on metabolomics can theoretically improve care?
Given the extensive HF metabolomics studies previously published (ref 13) in over 500 patients how does this manuscript advance the field?
The samples for the current study were collected in 2012 which so
What do the authors mean by HF-related death? And why were HF-related death and not cardiovascular death chosen?
The authors should explain why there is very little resemblance of metabolites in this study predicting outcome in HF pts as compared their previously published study.
Also the authors indicted decatrienoylcarnitine and phenylacetylglutamine were selected given their strong prediction capabilities yet they have chosen , paraxanthine, therobromine, and theophylline to be verified by a targeted approach.
For this study the authors should implement a targeted approach for decatrienoylcarnitine and phenylacetylglutamine and not for paraxanthine, therobromine, and theophylline.
The algorithm utilized should be utilized in another HF cohort to conform its validity.
What is the benefit of searching non endogenous databases since it does not improve understanding the pathophysiology of heart failure. As an example, how does caffeine/Theophylin levels improve our understanding of HF pathophysiology. The authors have made a valiant effort in trying to explain the impact of caffeine, but goes to show that finding mass spectrometric features in plasma that change in disease state is not enough to impact understanding of pathophysiology.
Reviewer 3 Report
This study is aiming to identify metabolites in plasma with high prediction power for heart failure (HF) related events using untargeted metabolomics approach. The justification for the study is to improve HF-patient outcomes by increasing knowledge in pathophysiological changes in metabolic profile. The text provides are valid rationale for the study. Study design is explained in great detail.
When performing multiple t-tests, p-values need to be corrected for the inflated Type I error rate. Has this been done for Table 1 and Figure 2C?